# The use of technology for social interaction by people with dementia: A scoping review

**Merryn Anderson**[1]*, **Rachel Menon**[2], **Katy Oak**[3], **Louise Allan**[4]

**1** College of Medicine and Health, University of Exeter, Exeter, United Kingdom, **2** Cornwall Partnership NHS Foundation Trust, Bodmin, United Kingdom, **3** Knowledge Spa, Royal Cornwall Hospital Trust, Truro, United Kingdom, **4** Centre for Research into Ageing and Cognitive Health, College of Medicine and Health, University of Exeter, Exeter, United Kingdom

* M.Anderson8@exeter.ac.uk

## Abstract

People with dementia (PwD) are at risk of experiencing loneliness, which is associated with physical and mental health difficulties [1]. Technology is a possible tool to increase social connection and reduce loneliness. This scoping review aims to examine the current evidence regarding the use of technology to reduce loneliness in PwD. A scoping review was carried out. Medline, PsychINFO, Embase, CINAHL, Cochrane database, NHS Evidence, Trials register, Open Grey, ACM Digital Library and IEEE Xplore were searched in April 2021. A sensitive search strategy was constructed using combinations of free text and thesaurus terms to retrieve articles about dementia, technology and social-interaction. Predefined inclusion and exclusion criteria were used. Paper quality was assessed using the Mixed Methods Appraisal Tool (MMAT) and results reported according to PRISMA guidelines [2,3]. 73 papers were identified publishing the results of 69 studies. Technological interventions included robots, tablets/computers and other forms of technology. Methodologies were varied and limited synthesis was possible. There is some evidence that technology is a beneficial intervention to reduce loneliness. Important considerations include personalisation and the context of the intervention. The current evidence is limited and variable; future research is warranted including studies with specific loneliness outcome measures, studies focusing on PwD living alone, and technology as part of intervention programmes.

## Author summary

More people are now living with dementia than ever before. People with dementia often experience loneliness. There has been increasing interest in using technology to help people with dementia connect with others and feel less lonely. Here we have searched for studies about people with dementia using technology for social interaction. We wanted to see what technologies are being used and if they are helpful or not. We found that there is a wide variety of types of technology being used to help social interaction for people with dementia. Types of technology included robots, tablet and desktop computers and a wide variety of other technologies. The studies we found used a diverse range of methods to see if the technology was helpful. Overall we found that technology could be a useful tool to

**Data Availability Statement:** All relevant results data is included in the manuscript. The search strategy is available on the Open Science Framework (DOI: 10.17605/OSF.IO/E7C2S).

**Funding:** LA is supported by the National Institute for Health Research Applied Research Collaboration South West Peninsula (arc-swp.nihr.ac.uk). MA is an Academic Clinical Fellow supported by the National Institute for Health Research (nihr.ac.uk). The views expressed in this publication are those of the authors and not necessarily those of the National Institute for Health Research or the Department of Health and Social Care. The funders had no role in study design, data collection and analysis, decision to publish, or preparation of the manuscript.

**Competing interests:** The authors have declared that no competing interests exist.

help reduce loneliness in people with dementia. However there needs to be more research into this area. Future research could focus on helping people with dementia who live alone, and using technology as one part of broader intervention programmes.

## Introduction

It is estimated that there are 885,000 people in the UK living with dementia; this is projected to increase to over 1.5 million people by 2040 [4]. Dementia has wide ranging consequences; van Wijngaarden, et al. investigated what it means to live with dementia; they found life could be isolating and some participants expressed feeling imprisoned at home [5]. This supports findings from the Alzheimer's society 2013 report: a third of PwD reported losing friends since diagnosis, 39% reported loneliness, increasing to 62% if they lived alone [6]. The impact of covid-19 has further negatively affected loneliness and mental health in PwD [7,8].

A scoping review by Courtin & Knapp looked at the relationship between loneliness and health in old age [1]. Of 128 studies included only two did not find a negative impact on health; consequences included increased risk of depression, increased risk of physical health conditions and negative impact on cognition. Loneliness is also associated with reduced quality of life overall [9].

Technology is used to connect with family, friends, and strangers all around the world. Although this has raised concerns regarding confidentiality and replacement of human care, it has the potential to be a tool to reduce loneliness in PwD. Studies have found that in the 'older adult' population increased internet usage is associated with reduced loneliness [10,11]. The systematic review by Brown & O'Connor into the use of mobile health applications by PwD found seven of nine studies had outcomes related to social health [12]. Mobile health applications stimulated conversation and facilitated intergenerational relationships. Focusing on the use of low-cost pet robots by PwD a scoping review found eight of the identified studies had outcomes related to communication/social interaction (SI) and that robots had an overall positive effect [13]. A systematic review including eighteen studies found that tablets, social robots, and computers have been used to support communication between PwD and their carers. They found that devises facilitated 'breaking the ice', increased interaction, facilitated understanding of the PwD and reduced pressure for the conversation partner [14].

The current body of evidence suggests that tablets, computers, and robot technologies are useful tools for PwD facilitating SI with people in the same location. However, this does not encapsulate other mediums of technology, nor does it provide information on the use of technology for distance communication. This scoping review uses a broad definition of technology and aims to look at the current evidence regarding the use of technology by PwD to facilitate SI.

## Methods

### Data sources

This paper utilises the Preferred Reporting Items for Systematic reviews and Meta-Analyses extension for Scoping Reviews (PRISMA-ScR; S1 PRISMA Checklist) guidance to provide the review structure [3]. A literature search was conducted using Medline, Cochrane database, NHS evidence, Trials registers, Open Grey, PsychINFO, Embase and CINAHL on 23rd April 2021. A sensitive search strategy was constructed using combinations of free text and thesaurus terms to retrieve articles about dementia, technology, and SI (S1 Table). An additional

amended search using the equivalent search terms was performed on ACM Digital Library and IEEE Xplore. The search was conducted by a specialist librarian (KO) and was registered with the Open Science Framework (DOI: 10.17605/OSF.IO/E7C2S). No limits were applied.

## Study selection

Titles and abstracts were screened for relevance and adherence to the inclusion criteria by one reviewer (MA), a random selection (10%) was screened by a second reviewer (RM) for comparison. Studies were included if they investigated the use of a technological device (e.g., tablet, robot etc.) by PwD and the study reported an outcome related to SI. Full texts were reviewed for exclusion, disagreements were resolved through consensus. Studies were excluded if they did not include primary data, if the population was not PwD, purpose of the technology was not SI, or if there were no outcomes related to SI. There were no exclusions related to study design. Although review articles without primary data were excluded from the results table, they were used to identify additional references.

Papers were assessed for quality using the Mixed Methods Appraisal Tool (MMAT), this tool allows studies using different methodologies to be compared. For each study type there are 5 specific criteria to allow quality assessment and comparison [2].

## Synthesis

Studies were grouped for comparison based on study methodology as defined by the MMAT [2]. Studies were sub-divided by technology type and outcome measure. A narrative approach was used to explore study results, identify themes, and provide comparison. Qualitative studies and Mixed Methods studies were read to identify commonalities in the emergent themes. These were then used to generate overarching themes related to the outcomes of this review.

## Results

The search identified 9161 papers (duplicates removed) of those 73 papers satisfied the inclusion/exclusion criteria. The PRISMA diagram is shown in Fig 1.

Of the 73 papers identified eight published results from four studies. Astell, et al. published two papers with results from the same participants using CIRCA [15,16]. Karlsson, et al. published two papers with results from the same participants using a digital photography activity diary [17,18]. D'Onofrio, et al. [19] and Casey, et al. [20] published results from a study using MARIO in residential care. To avoid over-representation of these studies the most recent papers have been included for analysis. Moyle, et al. published two papers with results from a study using Giraff in residential care [21,22]; the 2014 paper publishes more details of outcomes relevant to this review and is included in the analysis [21]. Three papers published results from more than one study Lancioni, et al. published results from two interventions [23], Huldgren, et al. published results from three interventions [24] and Smith published results from one intervention in two settings [25].

### Characteristics of the included studies

Key study information including design and methodology is summarised in Table 1. Studies were conducted in Asia, Europe, South America, North America, Australia and New Zealand. The interventions, study design and outcome measures are heterogeneous. Study setting was varied: 31 in residential care, 15 in participants' homes, 11 in day care, 3 in labs, 2 in hospital, 2 in community groups, 1 in a workshop and 8 used a mixture of settings. Proportion of studies in each setting is shown visually in Fig 2.

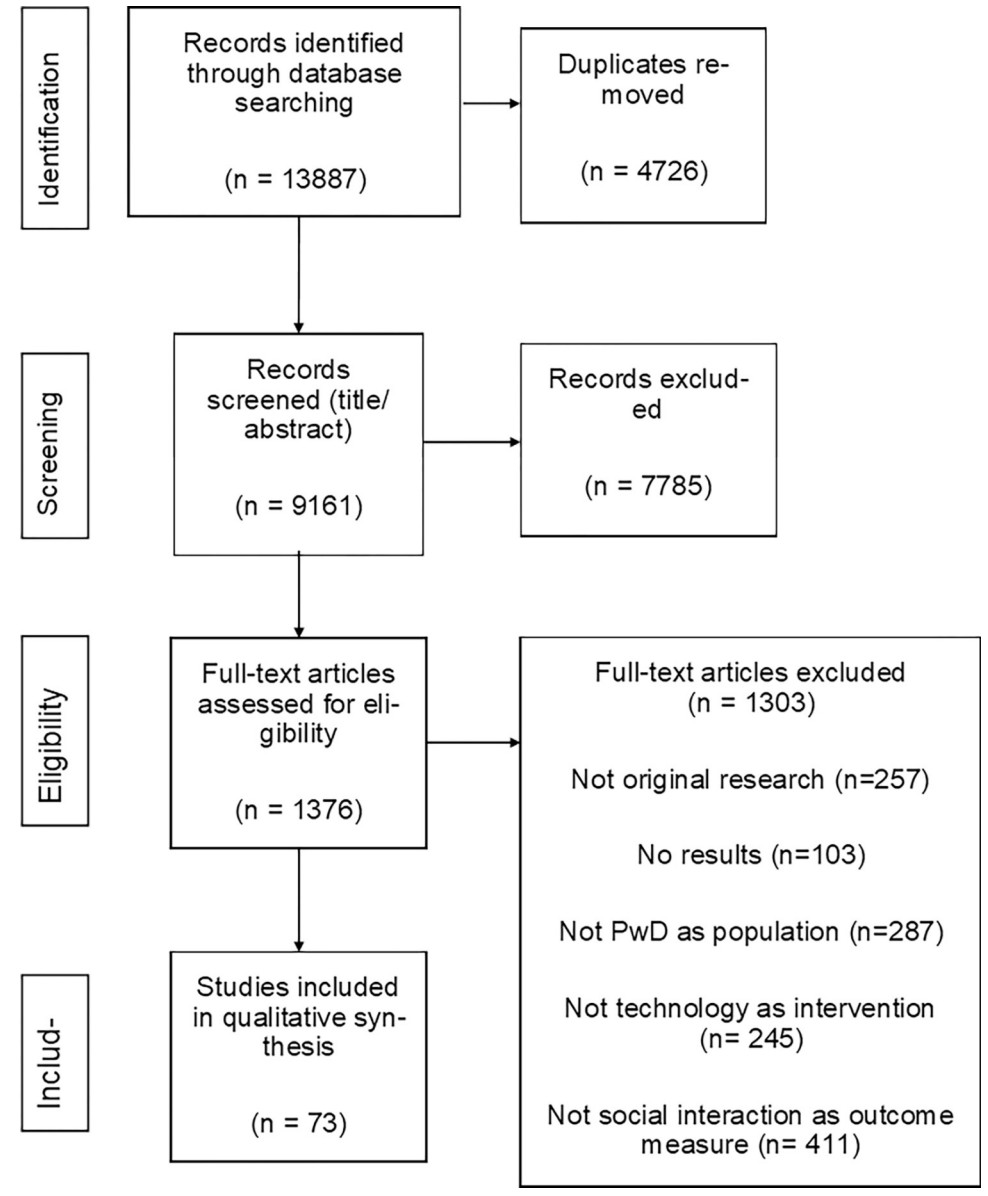

**Fig 1. PRISMA diagram.**

Three different clusters of technology type were identified: Robots, Computer/tablet programmes, and other forms of technology. An overview of broad technology type and main purpose of technological intervention is given in Fig 3.

## Robot based interventions

Studies have been grouped by robot type: social robots, pet robots and telepresence robots. A description of the different robots can be found in Table 2.

*Social robots.* Two studies used MARIO to prompt SI with the robot and other people [20,26]. Robinson, et al. compared Guide with Paro for impact on interaction with the robot and other people [27].

**Table 1. Overview of key study information including intervention, study design and methodology.**

| Paper | Continent | Intervention | Purpose of technology | Type of interaction studied | Comparison intervention | Setting technology used in | Characteristics of participants (people with MCI or dementia) | | | | Informal caregiver as participant? | Research Method (MMAT) | Quality Score (MMAT) |
|---|---|---|---|---|---|---|---|---|---|---|---|---|---|
| | | | | | | | Number (F,M) of PwD | Age: μ, range | Stage of dementia | Living Conditions | | | |
| ***Robot based interventions*** | | | | | | | | | | | | | |
| Barrett, et al. [26] | E | MARIO | SR | WT; FF (2 people) | Baseline | RH | 10 (7, 3) | 83,? | Mi-S | RH | No | NR-BA | 3 |
| Casey, et al. [20] | E | MARIO | SR | WT; FF (2 people) | None | H, RH, Ho | 38 (24, 14)* | 77, 55–93 | Mi-S | H, RH | Yes | Q | 5 |
| Robinson, et al. [27] | NZ | Guide | SR | WT; FF (2 people) | Paro | RH | 10 (5,5) | ?, 71–93 | ? | RH | Yes | MM | 2 |
| Chu, et al. [28] | Au | Sophie and Jack (NEC) | SR | WT; FF (group) | None | RH | 139 (44, 95) | ?, 65–90 | Mi-S | RH | No | QD | 3 |
| Khosla, et al. [29] | Au | Matilda (NEC) | SR | WT; FF (group) | None | RH | 115 (80, 35) | ?, 65–90 | Mi-S | RH | No | QD | 4 |
| Khosla, et al. [30] | Au | Betty (NEC) | SR | WT | None | H | 5 (?,?) | ?, 75–85 | ? | H (1+carer) | Yes | MM | 2 |
| Kase, et al. [31] | As | Telenoid R3 | SR | WT; FF (group) | Reminiscence | RH | 6 (?,?) | 87.5,? | MCI-Mo | RH | No | NR-CO | 3 |
| Chen, et al. [32] | As | Telenoid R4 | SR | WT | No robot | DC | 3 (2,1) | ?, 78–86 | Mi-Mo | H | No | MM | 1 |
| Kuwamura, et al. [33] | As | Telenoid R3b | SR | WT | No robot | RH | 3 (3,0) | ?, 85–96 | Mo-S | RH | No | NR-BA | 3 |
| Cruz-Sandoval & Favela [34] | SA | Eva robot | SR | WT | Robot using basic communication strategies | RH | 12 (?,?) | 80.25, 71–90 | Mi-Mo | RH | No | NR-BA | 2 |
| Pou-Prom, et al. [35] | NA | Milo R25 robot (Robokind; autonomous) | SR | WT | Human or Milo (Wizard-of-Oz) | RH | 19 (16, 3) | 88, 67–96 | Mi-S | RH | No | MM | 3 |
| Begum, et al. [36] | NA | Ed | Prompt tea making exercise | WT | None | Simulated Home | 10 (6, 4) | ?, 59–88 | Mi-S | Unknown | Yes | MM | 3 |
| Lima, et al. [37] | As | Hybrid face robot | SR | WT | None | Lab | 1 (0,1) | 67 | Mi | H | Yes | QD | 2 |
| Joranson, et al. [38] | E | Paro | PR-seal | WT; FF (group) | None | RH | 23 (16, 7) | 84.65, 62–92 | Mi-S | RH | No | QD | 3 |
| Liang, et al. [39] | NZ | Paro | PR-seal | WT; FF (2 people) | Standard care | H or DC | 30 (19, 11) | ?, 67–98 | ? | H | Yes | RCT | 3 |
| Song [40] | As | Paro | PR-seal | WT; FF (2 people) | No intervention | RH | Intervention 17 (17,0); Control 15 (15,0) | 83.94,?; 85.07,? | Mi-Mo | RH | No | NR-CO | 3 |
| Takayanagi, et al. [41] | As | Paro | PR-seal | WT; FF (2 people) | Stuffed toy | RH | Mild-mod group 25 (?,?); Severe group 11 (?,?) | 84.9,?; 87.5,? | Mi-S; S | RH | No | NR-CO | 2 |
| de 'Sant Anna, et al. [42] | NA | Paro | PR-seal | WT; FF (2 people) | None | RH | 5 (?,?) | ?, 66–96 | S | RH | No | Q | 2 |
| Hung [43] | NA | Paro | PR-seal | WT; FF (2 people) | None | Ho | 10 (4,6) | ?, 60+ | Mi-S | Unknown | No | Q | 4 |
| Kelly, et al. [44] | NA | Paro | PR-seal | WT | None | Ho | 55 (38,17) | 85.5, 67–104 | ? | Unknown | No | QD | 3 |
| In Soon & Hee Sun [45] | As | Paro | PR-seal | WT | Baseline | RH | 17 (17, 0) | 86.8,? | ? | RH | No | NR-BA | 3 |
| Shibata [46] | E, NA | Paro | PR-seal | WT; FF (group) | None | RH | Unknown | ?,? | Mi-S | RH | No | Q | 1 |
| Kramer, et al. [47] | NA | AIBO | PR-dog | WT; FF (2 people) | Human interaction, dog | RH | 18 (18, 0) | ?,? | ? | RH | No | NR-CO | 3 |
| Tamura, et al. [48] | As | AIBO (as robot and disguised as dog) | PR-dog | WT; FF (2 people) | Dog toy | RH | 13 (1,12) | 84,? | S | RH | No | NR-CO | 4 |
| Gustafsson, et al. [49] | E | JustoCat | PR-cat | WT; FF (2 people) | None | RH | 4 (2,2) | ?, 82–90 | S | RH | Yes | Q | 2 |

(*Continued*)

**Table 1.** (Continued)

| Paper | Continent | Intervention | Purpose of technology | Type of interaction studied | Comparison intervention | Setting technology used in | Characteristics of participants (people with MCI or dementia) | | | | Informal caregiver as participant? | Research Method (MMAT) | Quality Score (MMAT) |
|---|---|---|---|---|---|---|---|---|---|---|---|---|---|
| | | | | | | | Number (F,M) of PwD | Age: μ, range | Stage of dementia | Living Conditions | | | |
| Pike, et al. [50] | E | Ageless Innovation Robot Cat | PR-cat | WT; FF (2 people) | None | H | 12 (11,1) | ?,? | ? | H (inc sheltered accommodation) | Yes | Q | 4 |
| Feng, et al. [51] | E | LiveNature | PR-sheep and augmented reality display (ARD) | WT; FF (2 people) | Robot and ARD (off) or ARD (off) | RH | 16 (12,4) | 85.2,? | Mi-S | RH | No | RCT | 2 |
| Moyle, et al. [21] | Au | Giraff | Videoconferencing | Distance communication | None | RH | 5 (4,1) | ?, 79-89 | Mo | RH | Yes | MM | 4 |
| Moyle, et al. [52] | Au | Giraff | Videoconferencing | Distance communication | None | Lab | 5 (1,4) | 78.4, 69-87 | Mi-S | H | Yes | MM | 2 |
| *Computer or tablet based interventions* | | | | | | | | | | | | | |
| Astell, et al. [16] | E | CIRCA | Reminiscence | FF (2 people) | Reminiscence | DC or RH | 11 (6,5) | 83.54, 65-95 | Mi-S | H or RH | No | NR-CO | 3 |
| Alm, et al. [53] | E | CIRCA | Reminiscence | FF (2 people) | Reminiscence | DC or RH | 18 (13,5) | ?,? | Mo-S | H or RH | No | RCT | 2 |
| Purves, et al. [54] | NA | CIRCA | Reminiscence | FF (2 people) | None | RH | 3 (3,3) | ?, 81-90 | Mo | RH | Yes | Q | 2 |
| Samuelsson & Ekström [55] | E | CIRCA and CIRCUS | Reminiscence | FF (2 people) | None | RH | 3 (3,0) | ?,? | ? | RH | No | Q | 4 |
| Samuelsson, et al. [56] | E | CIRCA | Reminiscence | FF (2 people) | None | RH | 5 (3,2) | ?, 62-89 | Mi-S | RH | No | Q | 5 |
| Moon & Park [57] | As | Digital Reminiscence Therapy | Reminiscence | FF (2 people) | Storytelling | DC | Intervention 25 (25,0); Control 24 (24,0) | 82.96,?; 84.05,? | Mo | H | No | RCT | 2 |
| Pringle & Somerville [58] | Eu | CART Project | Reminiscence | FF (2 people) | Structured conversation or memory book | RH | 8 (?,?) | ?,? | ? | RH | No | Q | 1 |
| McAllister, et al. [59] | Au | Memory Keeper | Reminiscence | FF (2 people) | None | RH | 3 (1,2) | ?, 76-87 | ? | RH | Yes | Q | 2 |
| Yu, et al. [60] | NA | Memory Matters | Reminiscence | FF (2 people) | Wait list | H | 80 (46,34) | 82.1, 62-98 | Mi-S | H or RH | Yes | RCT | 4 |
| Dynes [61] | NA | Pictello App | Reminiscence | FF (2 people) | Baseline | H | 7 (2,5) | 69.7,? | Mi-Mo | H | Yes | NR-BA | 3 |
| Aitken [62] | E | Pictello App | Reminiscence | FF (2 people) | Baseline | H | 4 (1,3) | ?, 61-88 | Mi-S | H | Yes | NR-BA | 3 |
| Ekström, et al. [63] | E | GoTalk NOW | Conversation prompt | FF (2 people) | No tablet | H | 1 (1,0) | 52 | ? | H | Yes | MM | 2 |
| Tyack, et al. [64] | E | App with pictures of art | Conversation prompt | FF (2 people) | None | H | 12 (4,8) | 75, 64-90 | ? | H | Yes | Mixed methods | 3 |
| Lancioni, et al. [65] | E | Female face asking generic questions | Conversation prompt | WT | Blank screen and baseline | DC | 8 (7,1) | 83, 77-89 | Mo | H | No | NR-BA | 2 |
| Lancioni, et al. [66] | E | Female face asking generic questions | Conversation prompt | WT | No prompting and baseline | DC | 6 (3,3) | 84, 77-93 | Mo | H | No | NR-BA | 2 |
| | | Personalised video clips with questions/comments | Conversation prompt | WT | No prompting and baseline | DC | 10 (8,2) | 82, 70-92 | Mo | H | No | NR-BA | 2 |
| Lancioni, et al. [23] | E | Personalised video clips with questions/comments | Conversation prompt | WT | No prompting and baseline | DC | 8 (5,3) | 82, 73-96 | Mo | H | No | NR-BA | 3 |
| Ehret, et al. [67] | E | Tablet based memory game | Conversation prompt | FF (group) | None | DC | 14 (7,7) | ?, 76-91 | MCI-S | H | Yes | MM | 3 |
| Lazar, et al. [68] | NA | It's Never 2 Late | Conversation prompt | FF (2 people) | None | RH | 5 (4,1) | 87.8,? | Mo-S | RH | Yes | MM | 2 |
| Nordheim, et al. [69] | E | Tablet computer with variety of apps | Conversation prompt | FF (2 people) | None | RH | 14 (12,2) | ?, 62-104 | Mo-S | Rh | No | MM | 3 |

*(Continued)*

**Table 1.** (Continued)

| Paper | Continent | Intervention | Purpose of technology | Type of interaction studied | Comparison intervention | Setting technology used in | Characteristics of participants (people with MCI or dementia) | | | | Informal caregiver as participant? | Research Method (MMAT) | Quality Score (MMAT) |
|---|---|---|---|---|---|---|---|---|---|---|---|---|---|
| | | | | | | | Number (F,M) of PwD | Age: μ, range | Stage of dementia | Living Conditions | | | |
| Welsh, et al. [70] | E | Ticket to Talk | Conversation prompt | FF (2 people & group) | None | H or RH | 2 (1,1) | ?, >90y | ? | H or HR | Yes | Q | 4 |
| | | | | | | RH | 10 (?,?) | ?, 80-95 | Mo-S | RH | No | | |
| Upton, et al. [71] | E | iPad—variety of Apps | Social interaction | FF (2 people) | None | RH | Topic guided interview: 10 (10,0) | ?,? | ? | RH | No | Q | 3 |
| | | | | | | | Case study: 1 (1,0) | 87 | ? | RH | No | | |
| | | | | | | | Field observations: 149 (116,33) | ?,? | ? | RH | No | | |
| Park, et al. [72] | NA | WeVideo | Social interaction | FF (group) | None | Workshop | 7 (3,4) | ?,? | Mi | H | Yes | Q | 1 |
| Smith [25] | E | Tablet computer with variety of apps | Social interaction | FF (group) | None | DC | 12 (9,3) | ?, 70-92 | ? | Home | Yes | Q | 5 |
| | | Tablet computer with variety of apps | Social interaction | FF (2 people) | None | H | 10 (5,5) | ?, 73-89 | ? | H | Yes | Q | 5 |
| Howe, et al. [73] | E | CAREGIVERSPRO-MMD | Communication network | Distance communication | None | H | 37 (16, 21) | 70.41,? | ? | H | Yes | QD | 3 |
| Asghar, et al. [74] | As | Assisted Brotherhood Community (ABC) | Communication network | Distance communication | None | H | 8 (1,7) | 70.3,? | Mi | H | No | Q | 4 |
| Burdea, et al. [75] | NA | BrightBrainer | Computer game based training | FF (2 people) | Baseline | DC | 1 (0,1) | 51 | S | H | Yes | NR-BA | 2 |
| Beentjes, et al. [76] | E | FindMyApps program | Programme to identify relevant Apps | FF (2 people) | Tablet but no App | H | Intervention: 28 (12,16) | 72, 62-92 | MCI-Mi | H | Yes | RCT | 4 |
| | | | | | | | Control: 31 (11,20) | 72, 51-86 | MCI-Mi | H | Yes | | |
| **Other technological interventions** | | | | | | | | | | | | | |
| Damianakis, et al. [77] | NA | DVD based reminiscence | Reminiscence | FF (2 people) | None | H or RH | 12 (?,?) | ?, 60-95 | MCI-? | H or RH | Yes | Q | 4 |
| Huldtgren, et al. [24] | E | Interactive multimedia book based reminiscence | Reminiscence | FF (2 people) | None | RH | 8 (7,1) | ?, >80y | Mi-Mo | RH | Yes | Q | 2 |
| | | Reminiscence Map | Reminiscence | FF (2 people) | None | RH | 1 (1,0) | ?,? | ? | RH | No | Q | 1 |
| | | Chrono TV | Reminiscence | FF (group) | None | DC | 6 (0,6) | ?,? | Mi-Mo | H | No | Q | 1 |
| Olsen, et al. [78] | NA | Memory Lane Project—Vintage cabinet and TV playing music and video clips | Reminiscence | FF (group) | Other activities | DC | 15 (12,3) | 82, 76-94 | Mi-S | H | No | NR-BA | 2 |
| Nijhof, et al. [79] | E | The Chitchatters—Interactive multimedia objects | Reminiscence | FF (group) | Other activity | RH or DC | 10 (6,4) | 69, 52-86 | Mi-S | H or RH | No | MM | 2 |
| Subramaniam & Woods [80] | E | Digital Life Storybook—Personalised DVD for reminiscence | Reminiscence | FF (2 people) | None | RH | 6 (4,2) | 82, 73-89 | Mi-Mo | RH | Yes | MM | 3 |
| Coelho, et al. [81] | E | Virtual Reality reminiscence | Reminiscence | FF (2 people) | None | H | 9 (6,3) | 85.6,? | Mo-S | H | Yes | MM | 3 |
| Topo, et al. [82] | E | Picture Gramophone multimedia program & Editor | Reminiscence | FF (group) | None | RH | 23 (15,8) | ?, 60-89 | Mi-S | RH | No | MM | 2 |
| Karlsson, et al. [18] | E | Memory Lane Project—Digital photography activity diary | Reminiscence/Conversation prompt | FF (2 people) | None | H | 7 (?,?) | ?, 72-81 | Mi-Mo | H | Yes | Q | 3 |
| Fried-Oken, et al. [83] | NA | AAC devise with voice output | Conversation prompt | FF (2 people) | AAC devise without voice output | RH or H | 30 (23,7) | 74, 50-94 | Mo-S | RH or H | No | RCT | 3 |
| Johnson, et al. [84] | NA | Online forum | Social interaction | Distance communication | None | H | ? | ? | ? | H | No | Q | 4 |
| Hicks [85] | E | Commercial gaming technologies | Social interaction | FF (2 people) | None | Community group setting | 22 (0,22) | ?, 68-90 | ? | H | Yes | Q | 5 |

*(Continued)*

**Table 1.** (Continued)

| Paper | Continent | Intervention | Purpose of technology | Type of interaction studied | Comparison intervention | Setting technology used in | Characteristics of participants (people with MCI or dementia) | | | | Informal caregiver as participant? | Research Method (MMAT) | Quality Score (MMAT) |
|---|---|---|---|---|---|---|---|---|---|---|---|---|---|
| | | | | | | | Number (F,M) of PwD | Age: μ, range | Stage of dementia | Living Conditions | | | |
| Cutler, et al. [86] | E | Commercial gaming technologies | Conversation prompt (gaming) | FF (group) | None | Community group setting | 29 (18,11) | ?, 65–80 | ? | H (inc supported living) | Yes | Q | 5 |
| Topo, et al. [87] | E | Pictophone—Phone adapted with photos and stored numbers | Aid for making phone calls | Distance communication | None | H | 6 (0,6) | ?, 55–90 | Mi-Mo | H | Yes | Q | 2 |

Abbreviations: Unknown/not reported (?); Europe (E); New Zealand (NZ), Australia (Au); Asia (As); South America (SA); North America (NA); Social Robot (SR); Pet Robot (PR); With technology (WT); Face-to-face (FF); Home (H); Residential Care Home (RH); Hospital (Ho); Day Care (DC); Mild Cognitive impairment (MCI); Mild (Mi); Moderate (Mo), Severe (S); Qualitative (Q); Randomised Controlled Trial (RCT); Quantitative Descriptive (QD); Non-randomised study (NR); Before and after (BA); Cross-over (CO); Mixed Methods (MM)

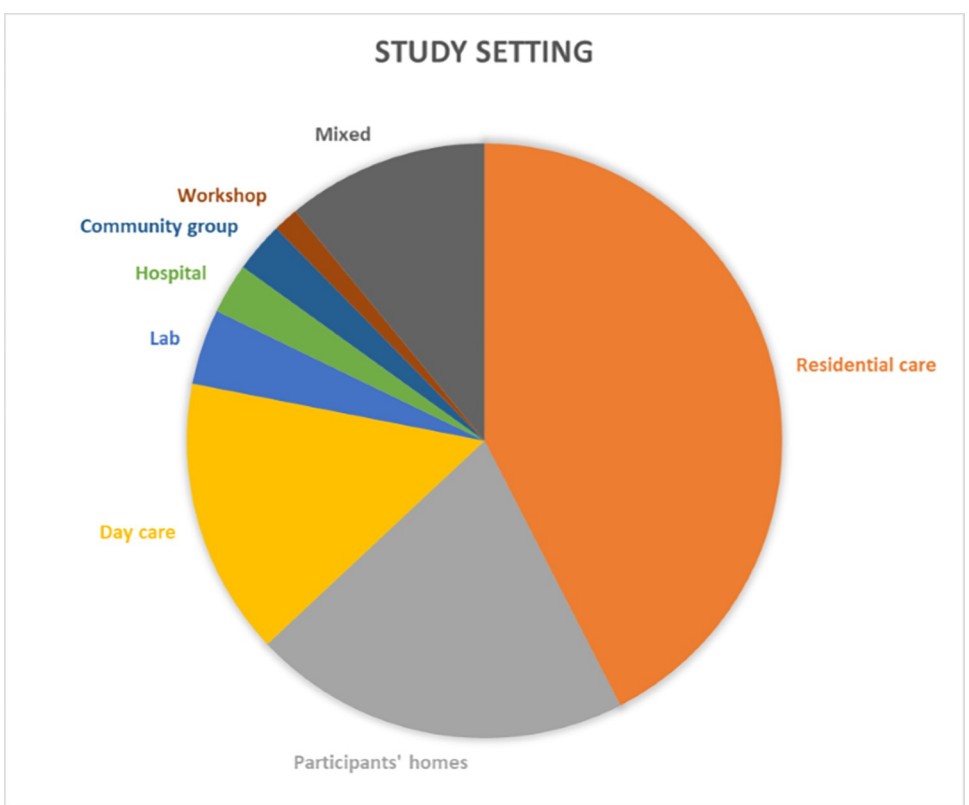

**Fig 2. Visual representation of proportion of studies in each study setting.**

Three studies used social robots from the Nippon Electric Company (NEC), two looked at SI with the robot and other people [28,29] and one looked at interaction with the robot alone [30].

Three studies used different versions of the Telenoid robot, one compared Telenoid facilitated reminiscence with traditional reminiscence, studying SI with the robot and within the group [31]. The other two studies investigated interaction with the robot alone [32,33]. Cruz-Sandoval & Favela investigated 'Eva's' ability to stimulate interaction using different communication strategies [34]. Pou-Prom, et al. compared the Milo R25 robot using autonomous speech to the same robot with a Wizard-of-Oz setup and human interaction [35]. Begum, et al. studied the use of an assistive robot for a tea-making exercise studying SI with the robot [36]. Lima, et al. studied the acceptability of the Hybrid Face Robot and reported results on interaction by the PwD with technology [37].

*Pet robots.* Nine studies used Paro; most of these studies looked at SI with the robot and other people prompted by the robot [38–43,46]. Two looked at interaction with the robot alone [44,45].

The two studies that used AIBO looked at interaction with AIBO and other people [47,48].

Two studies used cat like robots and studied interaction with the robot and others [49,50]. Feng, et al. used a sheep robot and investigated interaction with the robot and others [51].

*Telepresence robots.* Two studies used Giraff to facilitate SI with people in a different location to the PwD [21,52].

## Computer or tablet based interventions

The computer and tablet based interventions fell into different groups based on the purpose of the programme: reminiscence, conversation prompts, SI, communication networks and other.

## TYPE OF TECHNOLOGY AND PURPOSE

**Fig 3. An overview of broad technology type and main purpose of technological intervention.** Blues: Robot technologies, Greens: Tablet/computer based technologies, Oranges: Other technologies.

A description of the different types of computer and tablet based interventions can be found in Table 3.

Eleven studies used reminiscence programmes [16,53–62]. Nine used technologies as a conversation prompt [23,63–70]. Three used programmes to prompt SI. Park et al. used 'WeVideo' in a workshop format [72]. Upton, et al. looked at a variety of tablet based interventions that had already been rolled out into care settings and investigated the impact on SI [71]. Smith presents two studies in her PhD thesis investigating the use of a variety of Apps by PwD in Day Care and home settings [25]. Two of the studies used technology for communication with other people in a different location to the PwD [73,74]. Burdea, et al. used BrightBrainer[TM] and had an outcome of SI as reported by carers [75]. Beentjes, et al. included SI as an outcome for participants using the FindMyApps programme [76].

### Other forms of technology

Some studies used forms of technology that do not fall into the previous groups. They ranged from basic interventions such as a phone with pictures [87] to virtual reality [81]. The main

**Table 2. Description of robot technologies.**

| Robot name | Description |
|---|---|
| MARIO [20,26] | Robot with touch screen computer, also voice activated to allow two-way communications. |
| Guide [27] | Touch screen computer and verbal communication. |
| Social robots from the Nippon Electric Company (NEC) [28–30] | Robots are designed to deliver/participate in verbal and non-verbal communication, and they can also lead games, play music and dance. |
| Telenoid [31–33] | Humanoid robot using a Wizard-of-Oz system where a remote human operator controls the robot. |
| Eva [34] | Non-humanoid robot using Wizard-of-Oz system. |
| Milo R25 robot (Robokind) [35] | Humanoid robot with option of autonomous speech or Wizard-of-Oz system. |
| Hybrid Face Robot [37] | Affective hybrid face displayed on a tablet using Wizard-of-Oz system (has capacity to use Intelligent Virtual Assistant technology). |
| Paro [38–46] | Seal robot that can respond to interaction by moving and making noises. |
| AIBO [47,48] | Dog like robot, can follow set commands and non-verbally responds to speech/touch. |
| JustoCat [49] | Plush cat like robot that can respond to interaction by moving or making noises |
| Ageless Innovation robot cat [50] | Cat like robot which responds with movement and noises to light and touch |
| Sheep robot [51] | The robot could respond with sounds and movements, this was augmented with an interactive nature display |
| Giraff [21,52] | Telepresence robot, allows videoconferencing and can be controlled remotely to move around the PwD's living space. |

focus of these studies was reminiscence, although two of the studies used gaming technology to prompt conversation in group settings [85,86].

## Quality of included studies

Key methodological problems in the identified studies were: unclear research question, brief/poor reporting of methodology, limited explanation of data analysis, small number of

**Table 3. Description of Tablet and Computer based interventions.**

| Intervention | Description |
|---|---|
| CIRCA [16,53–56] | CIRCA is based on a touch screen computer and allows PwD and their cares to choose from photos, videos and pieces of music with an aim to prompt reminiscence. |
| Pictello App [61,62] | PwD and their carers can upload photos and audio recordings to the programme which can then be viewed to prompt reminiscence. |
| GoTalk NOW [63] | Designed for people with communication difficulties and is personalised to include multimedia both to prompt wide ranging conversations from current affairs to reminiscence and future plans. |
| Ticket to Talk [70] | An app designed to help younger people generate ideas and prompts to help facilitate conversations with PwD. |
| WeVideo [72] | Video editing programme used to create digital stories. |
| CAREGIVERSPRO-MMD [73] | Online chat and support forum which includes groups specifically for PwD to chat with their peers. |
| Assisted Brotherhood Community (ABC) project [74] | Connects PwD with others in their community to facilitate social interaction and provide informal support. |
| BrightBrainer™ [75] | Computer game based training programme. |
| FindMyApps [76] | App that helps PwD and their carers identify other Apps that might be beneficial/relevant to them. |

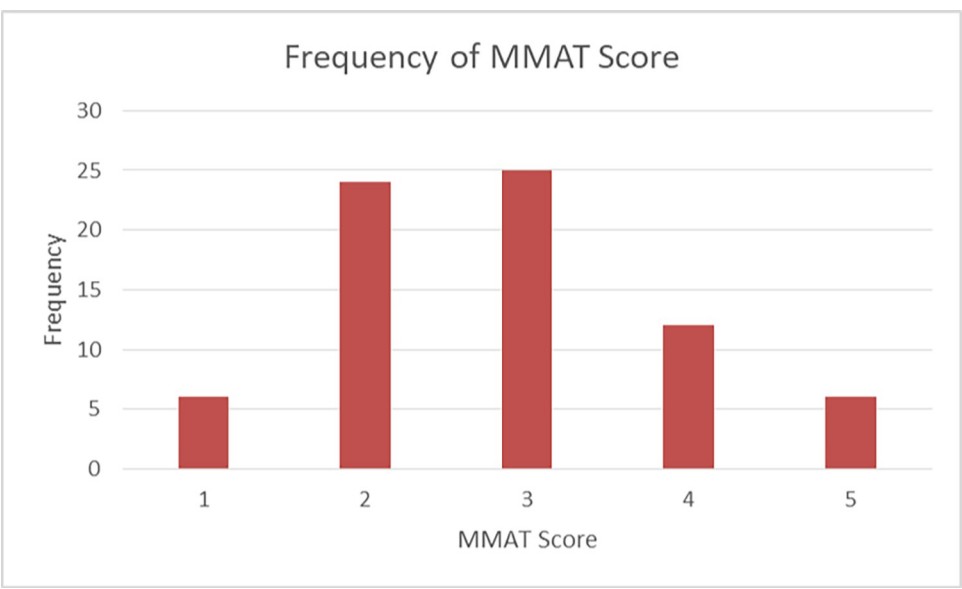

**Fig 4. Frequency of MMAT Scores.**

participants, participants were subset of a larger study, multiple outcome measures, and lack of accounting for confounders. The MMAT was used to assess papers for quality and risk of bias [2], details of MMAT score can be found in Table 1 and Fig 4 shows the frequency of each score. Most papers had a score of three or less indicating that the studies are limited by the quality of the methodology and risk of bias.

## Results of included studies

### Participants

Studies were limited by small participant numbers, and some had no justification for this (e.g. power calculation). Many of the studies were pilot or feasibility studies with an aim of investigating acceptability, usability, and functionality of the technology before further full-scale studies were carried out.

### Study type, outcome measures related to SI and results

There was a wide variety of study type in the papers found. To aid comparison, the papers have been grouped according to the study type and subdivided by technology. There was insufficient homogeneity in outcome measure to combine analyses.

### Qualitative studies

Twenty-three of the studies used qualitative methodology. Two papers included quantitative measurements in their study, but as they were not related to SI these papers have been included in this section [42,49]. An overview of the specific qualitative methodology and outcomes are shown in Table 4.

Six of the qualitative studies investigated the use of robots; one used a social robot and five used pet robots. The themes related to SI identified in the studies were conceptually similar and can be grouped into three broad themes: 'Relationships with the robot', 'Conversation

**Table 4. Overview of qualitative papers including methodology and outcomes.**

| Study | Intervention | Data collection | Emergent themes—relevant to review |
|---|---|---|---|
| Casey, et al. [20] | MARIO | SSI | Perceptions of MARIO; Impact of MARIO; Challenges in the use of social robots in the real-world context of dementia care |
| Hung [43] | Paro | Ob, FG, In | It's like a buddy': The robot helps people with dementia uphold or reclaim a sense of self in the world; 'It's a conversation piece': The baby seal facilitates social connection |
| de 'Sant Anna, et al. [42] | Paro | Ob | Communication occurs, and positive relationship with seal. But negative emotions at end of session. |
| Shibata [46] | Paro | Ob | Improved Communication; Improved Sociability |
| Gustafsson, et al. [49] | JustoCat | SSI | Interaction; Communication |
| Pike, et al. [50] | Ageless Innovation Robot Cat | In—multiple case study | Distraction; Communication; Connecting with the cat and connecting with others |
| Samuelsson & Ekström [55] | CIRCA, CIRCUS and no technology | Ob, SSI | Topic transitions; Initiatives; Maintaining conversation |
| Samuelsson, et al. [56] | CIRCA | In, SSI | Perceptions of today's experience; Perceptions of the conversation in presence of the tablet; Perceptions of the group |
| Purves, et al. [54] | CIRCA | Ob | Influence of program content on social interaction; Influence of program format on social interaction |
| Pringle & Somerville [58] | CART Project | Ob (by carer) | No themes reported but noted that the technology was observed to expand conversation |
| McAllister, et al. [59] | Memory Keeper | Ob, FG, In | Experienced and expected benefits of Memory Keeper; Engagement with and response to Memory Keeper by person with dementia |
| Damianakis, et al. [77] | DVD based reminiscence | Ob, In | Enhanced Communication and Leaving a Legacy |
| Huldtgren, et al. [24] | Interactive multimedia book based reminiscence | Ob, FG (with carers) | The book as a medium to support reminiscence; The book as a medium to support communication; Styles of leading the communication with the book; Accounting for individuality |
| | Reminiscence Map | Ob, FG | Triggers of memories; Communication pointers for others; Reciprocal communication |
| | Chrono TV | Ob | Reactions while viewing |
| Karlsson, et al. [18] | Memory Lane Project | SSI | Manifestations of Sense of Self; Sense of Self in Relation to Others |
| Park, et al. [72] | WeVideo | Ob | None |
| Johnson, et al. [84] | Online forum | Original posts on existing forum | Emotional; Informational; Companionship; Other |
| Hicks [85] | Commercial gaming technologies | Ob, In, FG | An opportunity to engage within the rural environment; Technology as an enabler |
| Cutler, et al. [86] | Commercial gaming technologies | Ob, Qu, FG | Promoting Lifelong Learning; Optimizing Mental, Physical, and Social Stimulation |
| Upton, et al. [71] | iPad—variety of Apps | SSI, Ob | Enhancing quality of life through touchscreen technology; Increasing Interpersonal Interactions; Inter-generational parity; Touchscreen technology as a challenge |
| Smith [25] | iPad—variety of Apps (day centre) | Ob | Technology interaction; Scaffolding and Support; Observed gains and limitations |
| | iPad—variety of Apps (home) | Ob, SSI | Expressed gains and limitations; Preferred activities |

*(Continued)*

**Table 4.** (Continued)

| Study | Intervention | Data collection | Emergent themes—relevant to review |
|-------|-------------|-----------------|-----------------------------------|
| Welsh, et al. [70] | Ticket to Talk | SSI | Promoting and Managing Reminiscence; Starting and Maintaining Conversation; Redistributing Agency |
| Asghar, et al. [74] | Assisted Brotherhood Community (ABC) | SSI | Needs Support; Social Support |
| Topo, et al. [87] | Pictophone | Qu, I | Communication via the phone |

Abbreviations: Semi-structured interview (SSI); Interview (In); Observation (Ob); Focus Group (FG); Questionnaire (Qu)

point' and 'Concerns'. A summary of the specific themes identified in each paper can be found in Table 5.

All six of the studies had themes encompassing 'Relationships with the robot'. PwD referred to robots as friends [20] and demonstrated a sense of emotional connectedness [43,50]. Pet robots led to a sense of purpose [50]. Communication was prompted and participants could speak to the robot in ways they couldn't with other people [42,46].

The five studies using pet robots had themes or concepts that fell into the overarching theme 'Conversation Point' and noted that participants spoke to the robot and to others about the robot [42,43,46,49,50].

Five of the studies had themes or concepts that can be grouped under the heading 'Concerns'. Some participants had a negative reaction to the robot [42,46,50]. A concern was raised that robots could be seen as a replacement for human interaction [20]. Gustafsson et al. found that the sense of responsibility for the robot could be too much [49].

Nine of the qualitative studies used technology for reminiscence. The themes related to SI identified in the studies were conceptually similar and can be grouped into the three broad themes; 'Communication prompt', 'Relationship facilitator' and 'Considerations'. A summary of the specific themes identified in each paper can be found in Table 6.

All the studies had themes that fall under the broad theme 'Communication prompt'; the media presented was noted to directly prompt communication by PwD in all the studies [18,24,54–56,58,59,70,77]. Three studies found that technology prompted conversation that was more PwD led [24,54,55]. Three found the nature of conversation changed when media was personalised or relevant to individual PwD [18,54,55]. One study noted that technology was a particular benefit for prompting intergenerational communication [77].

Five of the studies had themes related to the broad theme 'Relationship facilitator'. Participating in a study was noted to be beneficial [56] and the media presented prompted social activity [59]. Other studies found that the media promoted relationships more generally and reduced the power imbalance between 'carer' and 'cared for' [24,70,77].

Considerations raised in these papers included that the type/content of media influenced outcome [18,24,54], as did the setting [24,54]. As with the pet robots, it was noted that that technological interventions were not for everyone [59]. However, the study by Welsh, et al. suggested challenging topics shouldn't be avoided and media should allow and encompass a full range of emotions [70].

Five of the papers used technology as equipment for a shared activity which encouraged SI. The relevant themes were conceptually similar and can be grouped into the three broad themes; 'Communication Prompt', 'Relationship facilitator' and 'Considerations'. A summary of the specific themes identified in each paper can be found in Table 7.

**Table 5. Qualitative studies investigating robots–exploration of themes emerging.**

| | Grouping of themes identified by papers | | |
|---|---|---|---|
| **Paper** | **Relationship with the robot** | **Conversation point** | **Concerns** |
| Casey, et al. [20] | *'Perceptions of MARIO'*: PwD—robot as a friend, spoke about having a relationship with the robot. Carers—benefit of companionship *'Impact of MARIO'*: reduced loneliness and social isolation, had potential to increase connectivity. | | *'Challenges to the Use of Social Robots in the Real-World Context of Dementia Care'*: carers expressed concerned that MARIO could be seen as a replacement for human interaction. |
| Hung [43] | *"It's like a buddy': The robot helps people with dementia uphold or reclaim a sense of self in the world'*: benefit of emotional connectedness facilitated by 'non-verbal communication' by Paro. | *"It's a conversation piece': The baby seal facilitates social connection'*: Paro facilitated social connection both directly to Paro and also by mediating social connection with the facilitator. | |
| de Sant 'Anna, et al. [42] | Participants spoke to Paro; one participant's speech became clearer when with Paro. | One participant who was previously uncommunicative started initiating conversation when using Paro. | Negative feelings were expressed when session ended. One participant declined to participate in the intervention following the first session. |
| Shibata [46] | *'Improved communication'*: Paro enabled openness | *'Improved communication'*: prompted conversation about participants' past | *'Improved sociability'*: On seeing Paro one participant left the group stating "stupid thing" |
| | *'Improved sociability'*: one participant connected Paro with their pet dog | | |
| Gustafsson, et al. [49] | *'Interaction'*: sense of joy from interacting with JustoCat–it was "tolerant to love", "spoken about as a real cat". | *'Interaction'*: opening/prompt for conversation | *'Communication'*: PwD became worried about the cat–too much responsibility |
| | | *'Communication'*: common ground for communication | |
| Pike, et al. [50] | *'Distraction'*: some participants treated the robot cat like a real cat and formed a relationship with it. | *'Communication'*: the robot cat prompted conversations between PwD and their carers. | *'Connecting with the cat and connecting with others'*: some of the participants found the cats meowing distressing and wished for it to be turned off–emotional connection but detrimental to the participant |
| | *'Connecting with the cat and connecting with others'*: the cat gave some participants a sense of purpose as they had to care for the cat–a deeper emotional connection | | |

All the papers had themes encompassed by 'Communication Prompt' where technology facilitated communication between PwD and others [25,71,72,85,86]. All but one of the studies had outcomes that fall under the theme 'Relationship facilitator'. The shared activity helped provide a scaffold for SI and overcame barriers [71,85,86]. When PwD were interacting with those without dementia technology promoted a partnership rather than a 'teacher and student' relationship [25].

All of these papers had themes that encompassed 'Considerations'. Personalisation was found to be important to maximise engagement [85]. Some papers found that if the technology was too far outside a PwD's 'comfort zone' they engaged less [25,85,86], however PwD welcomed new experiences [86]. There were problems with equipment such as weight, connectivity [71] and usability [72].

**Table 6. Qualitative studies investigating the use of technologies to aid reminiscence–exploration of themes emerging.**

| Paper | Grouping of themes identified by papers | | |
| --- | --- | --- | --- |
| | Communication prompt | Relationship facilitator | Considerations |
| Samuelsson & Ekström [55] | Using CIRCUS, PwD led the most topic transitions, led and maintained the conversation more. | | |
| Samuelsson, et al. [56] | *'Perceptions of the conversation in presence of the tablet'*: CIRCA provided a conversation prompt but it was the group conversation that kept interest. | *'Perceptions of today's experience'*: experience of togetherness within the group and enjoyment from being with others.<br><br>*'Perceptions of the group'*: the group itself was seen as positive | |
| Purves, et al. [54] | *'Influence of program content on social interaction'*: Photos prompted conversations; nature of conversation determined by personal relevance.<br><br>*'Influence of program format on social interaction'*: Increased control by PwD and written information aided conversation maintenance | | *'Influence of program format on social interaction'*: Video format led to less conversation. The seating arrangement was important to allow engagement with the technology and eye contact between PwD and carers. |
| Pringle & Somerville [58] | Technology expanded conversation and increased the depth of PwDs' recollection. | | |
| McAllister, et al. [59] | *'Experienced and expected benefits of Memory Keeper'*: conversation prompt including increasing duration of connection/communication | *'Experienced and expected benefits of Memory Keeper'*: supported relationships and met emotional needs<br><br>*'Engagement with and response to Memory Keeper by person with dementia'*: positive memories triggered action e.g. dancing | *'Engagement with and response to Memory Keeper by person with dementia'*: one participant found photos of their family confusing. |
| Damianakis, et al. [77] | *'Enhanced Communication and Leaving a Legacy'*: DVD facilitated intergenerational communication | *'Enhanced Communication and Leaving a Legacy'*: DVD prompted deeper exploration of events | |
| Huldtgren, et al. Interactive Multimedia book [24] | *'The book as a medium to support reminiscence'*: generic and personal narratives by the PwD were prompted<br><br>*'The book as a medium to support communication'*: caregivers reported that the book was an aid to communication,<br><br>*'Styles of leading the communication with the book'*: the book led to question asking and more natural conversation | *'The book as a medium to support communication'*: caregivers noted that the book facilitated them learning something new about the PwD.<br><br>*'Styles of leading the communication with the book'*: equal turn taking and playfulness in the interaction was observed | *'Accounting for individuality'*: Carers' tailored the way they used the book as a tool based on their prior knowledge of the PwD |

*(Continued)*

**Table 6.** (Continued)

| Paper | Grouping of themes identified by papers | | |
| | Communication prompt | Relationship facilitator | Considerations |
|---|---|---|---|
| Huldtgren, et al. Reminiscence map [24] | *'Triggers of memories'*: story telling was prompted | *'Reciprocal communication'*: easy availability of communication prompt helped reciprocity in relationship | *'Communication pointers for others'*: could prompt communication between PwD |
| | *'Communication pointers for others'*: the map prompted others to start conversations | | |
| Huldtgren, et al. Chrono TV [24] | *'Reactions while viewing'*: PwD were quiet while viewing the TV | | *'Reactions while viewing'*: activity was passive |
| Karlsson, et al. [18] | *'Manifestations of Sense of Self'*: Photos prompted communication | | *'Sense of Self in Relation to Others'*: shared connection to a photo or photos of people prompted more in depth conversations. |
| Welsh, et al. [70] | *'Promoting and Managing Reminiscence'*: prompted 'comfortable' conversations | *'Promoting and Managing Reminiscence'*: conversations could become superficial if no common ground | *'Promoting and Managing Reminiscence'*: feedback that content should allow full range of emotions |
| | *'Starting and Maintaining Conversation'*: could lead to question-and-answer conversations | *'Redistributing Agency'*: reducing the power imbalance and allowing the PwD to lead improved relationships | |

Finally, three studies investigated unique technological interventions. Asghar, et al. studied a technologically mediated communication network aiming to link PwD with their neighbours. They found that as practical needs were met SI occurred [74]. Johnson, et al. investigated how PwD used an online support forum and found that post purpose could fall into four categories: emotional; informational; companionship; other [84]. Topo, et al. investigated the use of a modified telephone with stored numbers and picture prompts. They found the technology enabled independent call making, however technical issues made the phone difficult to use at times and a carer was needed to help with the phone programming [87].

## Quantitative Randomised Controlled Trials (RCT)

Seven of the papers used a RCT design. An overview of the outcome measures used, and results are shown in Table 8.

The two pet therapy studies had SI related outcomes that demonstrated a benefit of technology compared to control [39,51]. Technology facilitated reminiscence was not consistently better compared to controls [53,57,60]. The study using 'FindMyApps' did not find statistically significant benefits in SI related outcomes [76] and the study using a augmentative and alternative communication (AAC) device found voice output reduced conversation by the PwD [83].

## Quantitative Non-randomised Trials

*Before-and-after time series*: Eleven studies used a before-and-after time series design. One study used a non-randomised trial methodology, this study has been included in this section as the outcomes related to SI were only measured in the intervention arm [45]. An overview of the outcome measures used, and the results are shown in Table 9.

**Table 7. Qualitative studies investigating the use of technologies as equipment for a shared activity–exploration of themes emerging.**

| Paper | Grouping of themes identified by papers | | |
| --- | --- | --- | --- |
| | Communication prompt | Relationship facilitator | Considerations |
| Hicks [85] | *'Technology as an enabler'*: Individually tailored activities increased interest and prompted communication/ interaction. Technology acted as a scaffold for interaction | *'An opportunity to engage within the rural environment'*: opportunity for socialisation in a rural community. Technology provided a focus which made socialisation more relaxed. | *'Technology as an enabler'*: If individualisation couldn't happen participation was less. If games viewed as 'beyond [the individuals] capability' they were reluctant to participate |
| | | *'Technology as an enabler'*: the competitive nature of the games was valued by the participants and prompted friendly interaction | |
| Cutler, et al. [86] | *'Optimizing Mental, Physical, and Social Stimulation'*: novelty factor of new games stimulated light-hearted conversation. | *'Optimizing Mental, Physical, and Social Stimulation'*: Humour and discovery prompted team building. | *'Promoting Lifelong Learning'*: despite being unfamiliar with the technology participants were keen to learn |
| | | | *'Optimizing Mental, Physical, and Social Stimulation'*: games that were less physical were easier to engage with and found to be more enjoyable |
| Upton, et al. [71] | *'Enhancing quality of life through touchscreen technology'*: variety of apps encouraged individualised communication | *'Increasing Interpersonal Interactions'*: iPad in group and one-to-one settings increased interaction. | *'Touchscreen technology as a challenge'*: challenges such as weight and connectivity were noted |
| | | *'Inter-generational parity'*: iPads lead to increased inter-generational communication and collaboration | |
| Park, et al. [72] | Participants were willing to talk during the sessions. One participant commented that they enjoyed the social interaction of the group. | | Using the WeVideo programme was difficult for some participants and they needed facilitator support. |
| Smith (Day Centre) [25] | *'Technology interaction'*: the devise or App prompted conversation and the telling of anecdotes | *'Scaffolding and Support'*: tablet enabled partnership instead of 'teacher & student' interaction | *'Scaffolding and Support'*: carers and facilitators provided scaffold for PwDs' learning, if facilitator took 'expert' role this didn't work |
| | *'Observed gains and limitations'*: when participants achieved mastery of a game they improved in confidence and shared their achievement | *'Observed gains and limitations'*: technology led to chatting and laughter | *'Technology interaction'*: some participants weren't interested in the technology |
| | | | *'Observed gains and limitations'*: some participants were disengaged at times |
| Smith (Home) [25] | *'Preferred activities'*: Passive activities e.g. watching videos were a conversation prompt | *'Expressed gains and limitations'*: regular social contact beneficial | *'Expressed gains and limitations'*: in some dyads only the supporter gained while the PwD was disengaged |
| | | *'Preferred activities'*: objective of increased social contact met by sessions | |

**Table 8. Overview of papers using a Randomised Controlled Trial methodology including outcome measures and results.**

| Study | Intervention | Control | Outcome | Results |
|---|---|---|---|---|
| Liang, et al. [39] | Paro | Standard care | Custom observational tool—% of session when behaviours occurred (talk to others, talk to staff/activity coordinator, reciprocate, cooperate) | No significant difference in percentage of time spent talking to others, reciprocating or cooperating, increased percentage of time interacting with staff/activity coordinator in Paro group cc control (46.9% (SD 26.5) vs 25.5% (SD 24.3), p = 0.042) |
| Feng, et al. [51] | LiveNature (Sheep Robot and ARD) | Robot & ARD (off) or ARD (off) | Engagement of a Person with Dementia Scale (EPWDS) | Increased EPWDS composite sum in intervention cc control (p = 0.006) |
| Alm, et al. [53] | CIRCA | Reminiscence session | Custom observational tool—count (PwD choosing with and without prompt, caregiver providing prompts and conversation maintenance, both responding with memory, humour, laughter, or movement to music) | PwD chose more often with CIRCA (U = 2.00, p<0.001) and caregiver asked more direct questions with traditional session (U = 5.00, p = 0.01) |
| Moon & Park [57] | Digital Reminiscence Therapy | Storytelling session | Engagement of a Person with Dementia Scale (EPWDS) | No significant difference in EPWDS between digital and storytelling sessions. Statistically significant difference between mean difference in engagement between first and last session (p = 0.011). Digital session showed increased mean value of engagement between first and last session (3.78 +/- 3.82), whereas storytelling session showed decrease in mean value of engagement between first and last session (-0.86 +/- 6.01). |
| Yu, et al. [60] | Memory Matters (Individual and Group) | Wait list | Pleasant Events Schedule—AD | Individual MM had statistically significant better social interaction than group MM (P = 0.017) and control (P = 0.005) at six weeks but this was lost by 12 weeks |
| Fried-Oken [83] | Augmentative and alternative communication (AAC) Device: with voice output | AAC Device: without voice output | Utterances (counted) and coded to: topic maintenance, topic revival, topic elaboration or topic initiation. Also one word utterances and references to ACC devise. | More one word utterances (p<0.005), fewer total utterances (p<0.008) and fewer topic elaborations/initiations (p<0.004) when voice output present. |
| Beentjes, et al. [76] | FindMyApps program | Tablet but no App | Adult Social Care Outcomes Toolkit (ASCOT) and Maastricht Social Participation Profile (MSPP) | No significant difference in either measure between intervention and control |

**Table 9. Overview of papers using a Before and After Time Series methodology including outcome measures and results.**

| Study | Intervention | Comparison | Outcome | Results |
|---|---|---|---|---|
| Barrett, et al. [26] | MARIO | Baseline | Modified Observation, Multidimensional Scale of Perceived Social Support (MSPSS) | No statistically significant change in MSPSS score |
| Kuwamura, et al. [33] | Telenoid R3b | Face to face conversation | Custom questionnaire: assessing amount and quality of conversation. Completed by conversational participant (not PwD) and observer | No statistically significant difference in perceived amount or quality of conversation by conversational participant, nor amount of conversation perceived by observer. Statistically significant (p<0.01) difference quality of conversation as perceived by observer with better quality reported in face-to-face interaction. |
| Cruz-Sandoval & Favela [34] | Eva robot | Robot using basic conversational strategies | Custom observational tool: recording number of utterances and other behaviours/activities | Statistically significantly (p<0.05) increased number of utterances per minute for all participants (5/5) and number of sustained conversations for 4 participants when robot used sustained conversational strategies. |
| In Soon & Hee Sun [45] | Paro | Baseline | Observation table developed by Wada et al. | Statistically significant increase in total score for social interaction between pre and post test p < .001 |
| Dynes [61] | Pictello App | Baseline | Number of utterances for each code: Facilitation, Negotiation, Recognition, Validation | PwD and carers increased their use of person-centred communication strategies over the course of the intervention |
| Aitken [62] | Pictello App | Baseline | Number of on-topic utterances | No difference between baseline and treatment |
| Olsen, et al. [78] | Memory Lane Project - | Variety of alternative activities | Frequency of pre-determined behaviours during observation period | No impact on interaction cc controls |
| Lancioni, et al. [65] | Female face -generic questions | Blank screen and baseline | Frequency of micro switch activations and 'intervals' with verbal engagement/reminiscence | Increased micro switch activation and proportion of intervals with verbal engagement in intervention arm cc control arms |
| Lancioni, et al. [66] | Female face—generic questions | No prompting and baseline | Frequency of micro switch activations and 'intervals' with verbal engagement/reminiscence | Increased micro switch activation and proportion of intervals with verbal engagement in intervention arm cc controls |
| | Personalised video clips with questions/comments | No prompting and baseline | Frequency of micro switch activations and 'intervals' with verbal engagement/reminiscence | Increased micro switch activation and proportion of intervals with verbal engagement in intervention arm cc controls |

*(Continued)*

**Table 9.** (Continued)

| Study | Intervention | Comparison | Outcome | Results |
|---|---|---|---|---|
| Lancioni, et al. (Study 1) [23] | Personalised video clips with questions/ comments | No prompting and baseline | Frequency of micro switch activations, 'intervals' with verbal engagement/ reminiscence and computer reminders | Increased micro switch activation and proportion of intervals with verbal engagement in intervention arm cc controls |
| Burdea, et al. [75] | BrightBrainer | Baseline | Feedback questionnaires from informal care giver (Likert scale) | Improved verbal responses. From agree to strongly agree that subject was open to interact with others. |

Three studies used social robots and found varied results [26,33,34]. Only one of these studies found a benefit and it compared robots using different conversational strategies and did not compare robot to human interaction [34]. One study looked at the impact of a pet robot and found it increased SI [45]. Six looked at technology that promoted reminiscence and/or conversation. Two found a neutral effect when compared to baseline [62,78], and four reported positive effects [23,61,65,66]. One study looked at the impact of computer games on a PwD and found that the intervention had a positive effect on verbal responses and openness for SI [75].

*Cross-over design.* Six of the studies used a cross-over design. An overview of the outcome measures used, and the results are shown in Table 10.

The five robot interventions all demonstrated a negative or neutral effect of technology compared to control. These papers used counting methods to look at conversation [31,41,47,48] or social behaviour tools to rate SI [40]. The study utilising Telenoid R3 found PwD spoke more in a traditional reminiscence session compared to a robot facilitated session [31]. The other four robot studies were pet therapy models and compared robots to toys

**Table 10. Overview of papers using a Cross over methodology including outcome measures and results.**

| Study | Intervention | Control | Outcome | Results |
|---|---|---|---|---|
| Kase, et al. [31] | Telenoid R3 | Traditional reminiscence session | No. of utterances and sentence final particles | Half the participants had statistically significantly more utterances in traditional session cc telenoid |
| Takayanagi, et al. [41] | Paro | Stuffed toy | Time sampling method—count of talking/ utterances to toy/robot and to staff | Paro—Increased talking to robot in Paro session cc control (Mild/mod dementia group (p<0.01), severe dementia group (p<0.05). Decreased talking to staff in Paro session cc control in mild/mod dementia group (p<0.01). Decreased talking initiated by staff in Paro session cc control in mild/ mod dementia group (p<0.01) |
| Song [40] | Paro | No intervention | Social behaviour tools | No significant change in social behaviour outcome measures |
| Kramer, et al. [47] | AIBO | Human interaction, dog | Ethnologically derived categories: conversation, touch, looking at others, hand gestures, and smiles and laughs. | Statistically significantly fewer visitor initiation of conversation and participant response in AIBO group cc control. Statistically significantly more participant initiation of conversation in AIBO group cc control. Overall significantly fewer conversations in AIBO group cc control |
| Tamura, et al. [48] | AIBO (as robot and disguised as dog) | Dog toy | 6 categories: no interest, watching, talking, clapping hands, touching, and caring | Fewer episodes of interaction with AIBO cc toy dog overall (608 vs 985), including less talking (figures not available) |
| Astell, et al. [16] | CIRCA | Traditional reminiscence session | Verbal codes: PwD choosing with prompt, PwD initiation, Carer prompting, Carer conversation maintenance | CIRCA group PwD offered a choice more often cc trad group (t(10) = 5.9, p < .0005) and made more choices (t(10) = 3.617, p < .005; Table 3). More conversation maintenance in trad session (t(10) = 3.13, p < .01). Less initiation by PwD in trad session (z = 2.03, p < .05). |

[41,48], a real pet and human interaction [47] or no intervention [40]. Two of these studies found that the presence of the robots reduced communication overall [47,48] although Kramer, et al. found that conversation was initiated by PwD more in the robot group [47]. The study by Takayanagi, et al. found that in the robot group there was less talking between people but more spoken interaction with the robot when compared to the toy group [41]. Song's study found no significant change in the social behaviour outcome measures [40].

Astell, et al. used a cross-over design to look at the effect of CIRCA compared to a traditional reminiscence session [16]. They found that technology improved SI. PwD were offered and made more choices, initiated conversations more and carers used conversation maintenance techniques less.

## Quantitative descriptive studies

Six of the studies were non-comparative studies using a descriptive methodology. The study by Kelly, et al. was a before-and-after time series study however the data obtained relating to SI did not include any comparison between intervention/exposure and as such is quantitative descriptive data [44]. An overview of the outcome measures used (related to SI) and the results are shown in Table 11.

Two studies found Paro improved SI [38,44]. Three studies looked at social robots, using observation methods to count behaviours during the intervention. One found that interaction

**Table 11. Overview of papers using a Quantitative Descriptive Study methodology including outcome measures and results.**

| Study | Intervention | Outcome | Results |
|---|---|---|---|
| Kelly, et al. [44] | Paro | Modified coding schema based on study by McGlynn and colleagues: recording number of times specified behaviours occurred. | Speaking was the most commonly observed interaction occurring in 97% participants. 2/223 coded interactions were negative |
| Jøranson, et al. [38] | Paro | Observation of interaction with others and robot—week 2 and 10 | Conversation with Paro on the lap = 9% of the time (+/- 5.5), conversations without Paro on lap = 10.9% +/-10.0. Smile/laughter toward Paro 1.4% (+/-1.3), Smile/laughter toward other participants 0.8 (+/- 0.8) |
| Chu, et al. [28] | Sophie and Jack (NEC) | Custom Observation Scale | Interacting with robots increased from 2010 to 2014 (0.162, p<0.05). Interacting with others increased from 2010 to 2014 (0.152 p<0.05) |
| Khosla, et al. [29] | Matilda (NEC) | Observation of engagement scales adapted from other studies | No statistically significant change in verbal engagement measures. 60% participants responded that they liked participating in group activities with Matilda and 63% wanted Matilda to be their friend, neutral response to if helped make new friends. |
| Lima, et al. [37] | Hybrid face robot | Observational measure of engagement (OME) modified | No statistically significant results. Trend to longer duration of engagement from session 1 to 3. |
| Howe, et al. [73] | CAREGIVERSPRO-MMD | Data on use of platform | Median number of visits by PwD/6 months 29 (interquartile range = 114); 48.65% of PwD visited site < once a week; ~50% PwD did not do any social networking interactions |

with robots and others increased over time [28], one found no statistically significant change in verbal engagement with the robot over time, but questionnaire feedback response was positive [29] and the other study found no change over a shorter time period [37]. Finally, Howe, et al. investigated the impact of an online chat and support forum finding no benefit of the platform on SI for PwD [73].

## Mixed method studies

Sixteen of the studies utilised a mixed methods methodology. Studies were only included in this section if the quantitative and qualitative parts of the study had outcomes related to SI. The studies were subdivided by type of quantitative methodology used. There were no mixed methods studies that included a RCT. Nine studies included a before-and-after time series methodology [27,32,35,64,68,69,79,80,82]. One used a cross-over design [63]. Six used quantitative non comparative methodologies [21,29,36,52,67,81]. An overview of the outcome measures used (related to SI) and the results are shown in Table 12.

Most of the studies found either a neutral/mixed or positive impact of technology. Five of the studies looked at the impact of social robots [27,29,32,35,36]. Five looked at reminiscence technology or programmes [63,79–82]. Two investigated the benefits of movable videoconferencing technology [21,52]. Finally four used technology as a way to prompt SI and communication through games and other apps [64,67–69].

## Perspective of the Person with Dementia

Of the 69 papers included in this review 34 included the opinion of the PwD about the technology they had been using. PwD's perspective was included in 13 of the 29 robot studies, 13 of the 27 tablet/computer studies and 8 of the 13 studies using other forms of technology. Overall PwD enjoyed using technology however some found it difficult to use.

## Discussion

Having a diagnosis of dementia is associated with increased loneliness and social isolation; this has been worsened by the Covid-19 pandemic [5,6,8]. Technological innovations are one possible tool to alleviate loneliness and increase social connection, however their use is not without potential risks. This scoping review gives a comprehensive overview of the current available evidence related to the use of technology to benefit SI for PwD.

This review has shown that there is continued interest in PwD using technology to reduce feelings of loneliness and facilitate social connection. There is a variety of technological innovations that have been studied using various methodologies. Outcome measures are heterogeneous and limited comparison and synthesis has been possible.

### The impact of different technology types

This review presents weak evidence that robots reduce loneliness and/or increase social connectivity in PwD. The studies were frequently unclear regarding the intent of the intervention and what it was replacing or supplementing. When compared to person facilitated activities studies found negative or mixed results of robots [31,33,35,39,47], however there was evidence of more PwD led conversation with robots [32,47]. There were negative reactions to the robots reported in some studies [32,35]. However, if robot technology was going to be used it could be an addition rather than an alternative to human led activities. This might be more obvious in pet robots compared to social robots explaining why overall pet robots resulted in more positive outcomes than social robots. The study that compared a social robot to a pet robot found

**Table 12. Overview of papers using a Mixed Methods methodology including outcome measures, results and a summary of the relevant themes emerging from the qualitative data.**

| Study | Intervention | Comparison | Outcome (quantitative) | Specific qualitative data collection | Quantitative results | Qualitative results | | |
|---|---|---|---|---|---|---|---|---|
| | | | | | | Communication prompt | Relationship facilitator | Considerations |
| **Mixed methods including Before and After Time Series** | | | | | | | | |
| Robinson, et al. [27] | Guide | Paro | Behaviours (counting) | Semi-structured interview | No significant difference between interaction time with robot or with carer between two interventions. Statistically significantly more smiling, touching and speaking to Paro cc guide (all <0.05) | 'Overall Impression of Robots': personalisation of entertainment options on Guide helped prompt conversation | | 'Overall Impression of Robots': Robots not for everyone, Paro more suitable for PwD than Guide 'Improvements to Robots': Paro's noises could be distressing |
| Chen, et al. [32] | Telenoid R4 | No robot | Time participant and partner spent talking and time maintaining eye contact | Observation | Trend to increased participant/partner conversation ratio in family mediated robot session cc family session in 2/3 participants. 1/3 participant did not engage with robot at all. | | Tactile interaction with Telenoid occurred, it was treated like a baby | One participant found the robot distressing. One struggled to understand the robot's voice. |
| Pou-Prom, et al. [35] | Milo R25 robot (Robokind autonomous) | Human or Milo R25 robot (Wizard-of-Oz) | Utterances (counting) | Observation and questionnaire | Fewer utterances with robot (μ 8.41) cc Wizard-of-Oz (μ 15.5) and human (μ 22.5) (p<0.0001) | | 'Likeability of the robot': variable reactions, reluctance to engage by some | 'Understanding and Intelligibility': inhibited interaction especially in autonomous robot condition. 'Intelligence of the Robot': technological limitations of robot were noted 'Eliciting Reactions': people with higher MMSE interacted with the robot more easily. |
| Tyack, et al. [64] | App with pictures of art | None | Quality of Life-Alzheimer's Disease (QoL-AD) scale | Semi-structured interview | No significant change in wellbeing or quality of life across the intervention | | 'Dyad relationship': shared activity beneficial to relationship | 'Experience of app': some issues using App but overall good experience |
| Nijhof, et al. [79] | The Chitchatters | Non-tech game | Oshkosh Social Behaviour Coding (OSBC) scale (modified) | Semi-structured interviews | Most frequent behaviour type was social verbal behaviour (cc social non-verbal behaviour, non-social verbal/non-verbal behaviour). No difference in Social verbal behaviour nor Social non-verbal behaviour between people with low/high MMSE scores | 'The use of the CC in the daily work of activity facilitators': acted as a start point for conversations | 'Social behaviour of players with dementia': triggered shared memories which led to socialisation | 'Easy to use for players with dementia': PwD found the objects difficult to use. Format was less relevant for younger PwD. |
| Topo, et al. [82] | Picture Gramophone | None | Questionnaire including information on frequency of social contacts. Estimate of participation in activities and level of social interaction. | Semi-structured interview with PwD and staff members. Staff member journal entries | No statistically significant effect on social contacts or interaction. | Personalised images prompted reminiscence. | Music prompted social interaction including singing together and dancing. | PwD reported that they couldn't use the tech but then demonstrated that they could. |

(Continued)

**Table 12.** (Continued)

| Study | Intervention | Comparison | Outcome (quantitative) | Specific qualitative data collection | Quantitative results | Communication prompt | Qualitative results | |
|---|---|---|---|---|---|---|---|---|
| | | | | | | | Relationship facilitator | Considerations |
| Subramaniam & Woods [80] | Digital Life Storybook | None | Quality of life-Alzheimer's disease scale (QOL-AD); Quality of the caregiving relationship questionnaire24 (QCPR) | Semi-structured interviews | Improved average QOL-AD score and QCPR score but not documented if statistically significant. | 'Encourage conversation': prompted communication and interaction | 'Gained information and knowledge': knowing more about PwD's past helped develop relationships | |
| Nordheim, et al. [69] | Tablet with variety of apps | None | Quality of Life-Alzheimer's Disease (QoL-AD) scale | Semi-structured interviews, review of care records and observation | Minimal increase in QoL-AD score (baseline: 32.8 points; t1: 34 points; t2: 34.4 points) | 'Promote communication and interaction': communication prompt | 'Positive group dynamics': group working facilitated to relationship building. 'Reduction of neuropsychiatric Symptoms and other effects': improved mood and engagement | |
| Lazar, et al. [68] | It's Never 2 Late | None | Quality of Life-Alzheimer's Disease (QoL-AD) scale; positive affect instrument (PAI) | Semi-structured interviews | Some changes in QoL-AD seen but no statistical analysis performed. PAI baseline average score of 22.3 (SD 4.8) | 'Benefits': facilitated interactions, prompted reminiscence | 'Benefits': staff learnt more about residents; music bridged gap between generations | 'Challenges': technological issues–staff training needed. Nature of dementia could limit use. Concerns raised that technology could be a replacement for human care. 'Influencers': a facilitator was needed to allow the PwD to use the technology, one-to-one/small group was easier/better |
| Mixed methods including cross over design | | | | | | | | |
| Ekström, et al. [63] | GoTalk NOW | Interaction without tablet | Length of recording and observation | Observation | Increased mean length of recording with tablet cc without tablet (17.45 cc 6.05min). Initiatives/min by PwD 1.2 without tablet, 0.9 with tablet | Tablet resulted in 'do you remember/do you know' type questions | | Over course of intervention PwD learnt to use tablet independently |
| Non-comparative study | | | | | | | | |
| Khosla, et al. [30] | Betty (NEC) | None | Time spent on each function and questionnaire re perception of and reaction to robot | Observation of nature of interaction and engagement | Music/dance function was used the most and phone the least (no statistical comparison). From survey 4/5 PwD agree/strongly agree that Betty is a friend. | | Positive emotion was seen when PwD were dancing or singing with the robot | Unexpected responses from Betty led to negative emotions in the PwD. |
| Begum, et al. [36] | Ed | None | % of PwD who displayed each behaviour and frequency of behaviour | Observation, semi-structured interviews and questionnaires | 80% Participants verbally engaged with robot with an average of 7.1 times each. 10% initiated a conversation, average of 2 times each. 100% directed non-verbal cues to robot, average 15.1 times each. | | 'Trust': PwD would interact socially with robot but not turn to it for help. 'Communication: PwD appeared to enjoy communicating with the robot | 'Trust': PwD seemed more comfortable as the activity progressed suggesting familiarity was important. |
| Moyle, et al. [21] | Giraff | None | Number of and duration of calls and duration of engagement | Video recording, semi-structured interviews, research team observation and notes | Average engagement of 93% across calls | | 'Acceptability and implementation': reduced social isolation, video made connection easier than using a telephone. | 'Implementation and practicality': internet connection was problematic, unable to use wireless due to security concerns at care home |

(Continued)

**Table 12.** (Continued)

| Study | Intervention | Comparison | Outcome (quantitative) | Specific qualitative data collection | Quantitative results | Communication prompt | Qualitative results | | Considerations |
|---|---|---|---|---|---|---|---|---|---|
| | | | | | | | Relationship facilitator | | |
| Moyle, et al. [52] | Giraff | None | Observable Displays of Affect Scale; Modified-Temple Presence Inventory (Modified-TPI); International-Positive Affect, Negative Affect Scales–Short form (I-PANAS-SF) | Semi-structured interview | Modified-TPI: good-excellent presence factors, no statistically significant between perception of PwD, carers and family. I-PANAS-SF: significantly more positive affect than negative affect (p<0.001). ODAS trend to more positive than negative affect on subscales but only reached statistical significance on Facial Display subscale (p0.007) | | *'Understanding the social connection facilitated through the Giraff'*: video aspect allowed for greater immersion and *'realistic'* communication; manoeuvrability added to experience | | *'Understanding the social connection facilitated through the Giraff'*: manoeuvrability was a distraction from communication. *'Acceptability, satisfaction and attitudes'*: familiarisation improved acceptance. *'Utility'*: privacy concerns were raised by health care professionals |
| Coelho, et al. [81] | Virtual Reality reminiscence | None | Observation and Simulator Sickness Questionnaire | Observation and semi-structured interviews | Communication was spontaneous 57.7% of the time. Communication content was most frequently personal memories 71.1%, 56.2% of these were positive/happy. No cases of significant increase in simulator sickness symptoms. | *'Behaviour displayed during sessions'*: intervention prompted communication | *'Impact of the intervention'*: preparatory process of discussing ideas for immersive environment was beneficial to relationship with carers | | *'Behaviour displayed during sessions'*: some questions caused confusion. The immersive nature of the intervention had a transient benefit. |
| Ehret, et al. [67] | Tablet based memory game (standardised and individualised versions) | None | Observation (protocol) | Observation | Increased spontaneous communication in individualised mode. In standardised mode attention was on game not others. | *'Relationship between language and game'*: type of game changed the type of communication *'Observations in players with severe dementia'*: all games led to an element of story-telling and interaction with carer. | *'Relationship between language and game'*: some games led to socialisation with others becoming involved invoking a 'team spirit'. Element of helping others who were struggling to complete a game | | |

no significant difference in interaction time, but more smiling, touching and speaking to Paro compared to Guide [27].

The majority of papers utilising tablet or computer-based programmes found a positive effect; none found an overall negative effect. The most common purpose of intervention was to prompt reminiscence or conversation between people who were in the same place. A common finding was that the content or type of communication changed when a technological intervention was used compared to traditional conversation. Three studies found that the proportion of conversation led by PwD increased when using a tablet/computer [16,53,63]. Furthermore the content of the conversation changed, being more PwD led and reciprocal [16,53–55,58,61]. There was a theme that technology facilitated communication and collaboration between people of different generations, helping carers get to know the people they were caring for [68,71]. However, over-reliance on technology could lead to these conversations becoming superficial or 'question and answers' rather than reciprocal conversations [70].

The broad group of 'other' forms of technology allowed limited comparison. As this review was not limited by date of publication many of these interventions represent older technologies, however this does not make the results irrelevant to current practice. Simple solutions may be more accessible to PwD due to cost and familiarity. Coelho, et al. was the only paper found that used virtual reality (VR), they looked at SI between the PwD using VR and a conversation partner in the (real) room [81]. Although some may have concerns that technology such as VR may not be suitable for use by PwD this study found that it was beneficial and led to SI both during the intervention and in the preparatory sessions.

## Technology as an intervention to reduce loneliness and/or increase social connectivity

The 69 papers included in this study looked at 73 different interventions; of these 64 involved face-to-face interactions with other people in the same place. This suggests that the technology is being used as a facilitator for interaction that may have already been taking place. It also limits the applicability of the interventions to those who are at highest risk of loneliness, those who live alone or with limited opportunities to meet with others face-to-face. Thirteen of the interventions facilitated interaction with technology alone (without including face-to-face interaction with other people as part of the intervention) and six with other people who were in a different location to the PwD.

Another prominent theme of the studies found was that many included customisable or personalised interventions. Tablet interventions gave more benefit if the media was personally relevant [54,55]. The qualitative and mixed methods studies gave more insight into the importance of this. The study by Karlsson, et al. found two emergent themes related to this: 'Manifestations of Sense of Self' and 'Sense of Self in Relation to Others'. They noted that the degree of personal identification that a PwD felt with an image/media influenced how they responded to it, if both the PwD and their conversation partner identified with the image/media this led to the most in-depth conversations [18]. This was also true in 'off the shelf' games as studied by Hicks. In this study although the game content wasn't necessarily customisable the choice and content of the activity programme could be personalised. They found that individually tailored activities increased interest, communication, and interaction, whereas if individualisation couldn't happen the PwD participated less. Technology was viewed to be a scaffold for interaction, if it was too far outside the PwD's experience or comfort zone, they were more reluctant to engage [85].

A striking finding from this study is the proportion of papers that published the opinion of the PwD. Less than half of the papers included any subjective feedback directly from the PwD.

Although people with more severe dementia might struggle to remember previous sessions, they would often still be able to give an opinion during the session. Using a carer or family member's opinion is not a substitute for the PwD's opinion as they do not always have the same perceptions or experiences of dementia [88]. There is no reason why people with milder dementia would not be able to communicate their needs, how the robot met or failed to meet those needs or how the robot might be improved to enhance its efficacy.

## Strengths and limitations

This study has provided new insights into the breadth of technology that has been studied to improve SI or reduce loneliness in PwD. It provides a comprehensive overview of the current available evidence. It has highlighted the limited amount of data available in using technology to facilitate distance communication.

This study can only draw limited conclusions about the effectiveness of technological intervention for reducing loneliness/social isolation in PwD. It has been unable to generate any statistical comparison to allow robust conclusions to be made. This is due to the variability in outcome measures, heterogeneity in study design and comparison interventions. Many of the studies had multiple additional outcome measures and were not primarily designed to assess the impact on loneliness and/or social connection. None of the studies had a primary outcome measure that directly measured perceived loneliness. The studies also often lacked a clear 'real life' aim of how the intervention might be used to allow assessment of clinical/social efficacy. The quality of the interventions was variable as assessed by the MMAT [2].

## Future research

This review has identified multiple areas for future research. Homogeneity in outcome measures would be beneficial to aid comparison and allow meta-analysis. Increased focus on PwD at highest risk of loneliness or with reduced technology literacy would increase insights and improve clinical/social application. In particular focusing on PwD living in the community, and PwD who live alone or in more socially isolated locations would increase clinical/social relevance. Areas for future research regarding technology type include technology that facilitates social interaction between people in different locations and technology that is used as part of complex interventions to reduce loneliness and social isolation.

## Conclusion

Given the prevalence and impact of loneliness on PwD and the wide interest in using technology to help alleviate this it is important that there is robust research to investigate how best technology can be used. This needs to include the type of technology, the setting the technology is used in and clear outcome measures that provide meaningful data. There is less research looking at technological interventions connecting those in different locations which would be more applicable to those living alone and in rural communities and particularly pertinent given the Covid-19 pandemic.

There is evidence that technology could be a useful and beneficial intervention to help reduce loneliness and facilitate social connection. From the results found interventions that can be personalised and include some aspect of face-to-face intervention are promising. As a result of the lack of consistency between the studies available it is difficult to directly compare their results and generate conclusions that can translate into and inform clinical practice.

## Supporting information

**S1 PRISMA Checklist. Preferred Reporting Items for Systematic reviews and Meta-Analyses extension for Scoping Reviews (PRISMA-ScR) Checklist.**
(DOCX)

**S1 Table. EMBASE Search Strategy.**
(DOCX)

## Author Contributions

**Conceptualization:** Merryn Anderson, Louise Allan.

**Data curation:** Merryn Anderson, Rachel Menon, Katy Oak.

**Formal analysis:** Merryn Anderson.

**Investigation:** Merryn Anderson.

**Methodology:** Merryn Anderson, Rachel Menon, Katy Oak, Louise Allan.

**Supervision:** Louise Allan.

**Writing – original draft:** Merryn Anderson.

**Writing – review & editing:** Rachel Menon, Katy Oak, Louise Allan.

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
