## [Decision Letter · Decision Letter 0]

9 Feb 2022

PDIG-D-22-00008

The use of technology for social interaction by people with dementia: a scoping review

PLOS Digital Health

Dear Dr. Anderson,

Thank you for submitting your manuscript to PLOS Digital Health. After careful consideration, we feel that it has merit but does not fully meet PLOS Digital Health's publication criteria as it currently stands. Therefore, we invite you to submit a revised version of the manuscript that addresses the points raised during the review process.

The review is generally in a good shape. However, the reviewers suggested several clarifications/comments that may help you improve the paper. Please also consider expanding the search to technical repositories, such as IEEE Xplore and ACM Digital Library.

We look forward to receiving your revised manuscript.

Kind regards,

Shlomo Berkovsky

Section Editor

PLOS Digital Health

Journal Requirements:

1. Please amend your detailed Financial Disclosure statement. This is published with the article, therefore should be completed in full sentences and contain the exact wording you wish to be published.

i). Please include all sources of funding (financial or material support) for your study. List the grants (with grant number) or organizations (with url) that supported your study, including funding received from your institution. 

ii). State the initials, alongside each funding source, of each author to receive each grant.

iii). State what role the funders took in the study. If the funders had no role in your study, please state: “The funders had no role in study design, data collection and analysis, decision to publish, or preparation of the manuscript.”

iv). If any authors received a salary from any of your funders, please state which authors and which funders.

2. Please update your Competing Interests statement. If you have no competing interests to declare, please state: “The authors have declared that no competing interests exist.”

Additional Editor Comments (if provided):

Reviewers' comments:

Reviewer's Responses to Questions

**Comments to the Author**

1. Does this manuscript meet PLOS Digital Health’s publication criteria? Is the manuscript technically sound, and do the data support the conclusions? The manuscript must describe methodologically and ethically rigorous research with conclusions that are appropriately drawn based on the data presented.

Reviewer #1: Yes

Reviewer #2: Yes

2. Has the statistical analysis been performed appropriately and rigorously?

Reviewer #1: Yes

Reviewer #2: Yes

3. Have the authors made all data underlying the findings in their manuscript fully available (please refer to the Data Availability Statement at the start of the manuscript PDF file)?

Reviewer #1: No

Reviewer #2: Yes

4. Is the manuscript presented in an intelligible fashion and written in standard English?

Reviewer #1: No

Reviewer #2: Yes

5. Review Comments to the Author

Reviewer #1: This is an extensive survey paper to discuss how technology can be used for social interaction by people with dementia.

It would be good to include review comments in the following aspects.

1. Dataset. What will be dataset available (if any) in this area. How to collect data by researchers themselves.

2. Experimental setup and performance metrics. What will be standard protocol to conduct experiments and what metrics should be used for quantitative performance evaluation of various methods.

3. Suggestion for future study. Any insightful suggestions for future research studies?

4. Only two figures are provided in this 70-page paper, it is a bit dry. It would be good to include some figures to provide an overview of this research area.

Reviewer #2: This paper reports a scoping review of technology for increasing interaction for people with dementia, discusses its trends and highlights numerous potential works.

Overall the review is well performed, with a comprehensive review of the current literature. The review is also well written, balanced and clear. This paper identifies an essential gap in the current literature.

One comment:

The review only included articles indexed in medical databases as opposed to purely technical databases such as The Institute of Electrical and Electronics Engineers (IEEE) Xplore Digital Library. Considering that the focus of the review is on technology, this omission may weaken the impact of this paper. Please consider justifying within the text the reason for this choice.

6. PLOS authors have the option to publish the peer review history of their article (what does this mean?). If published, this will include your full peer review and any attached files.

**Do you want your identity to be public for this peer review?** For information about this choice, including consent withdrawal, please see our Privacy Policy.

Reviewer #1: No

Reviewer #2: No

---

## [Editor Report · Decision Letter 1]

25 Apr 2022

The use of technology for social interaction by people with dementia: a scoping review

PDIG-D-22-00008R1

Dear Dr Anderson,

We are pleased to inform you that your manuscript 'The use of technology for social interaction by people with dementia: a scoping review' has been provisionally accepted for publication in PLOS Digital Health.

Best regards,

Matthew Chua Chin Heng

Academic Editor

PLOS Digital Health